# Differential Coding of Fruit, Leaf, and Microbial Odours in the Brains of *Drosophila suzukii* and *Drosophila melanogaster*

**DOI:** 10.3390/insects16010084

**Published:** 2025-01-15

**Authors:** Claire Dumenil, Gülsüm Yildirim, Albrecht Haase

**Affiliations:** 1Centre for Mind/Brain Sciences (CIMeC), University of Trento, 38068 Rovereto, Italy; clairefrancenicolle.dumenil@unibz.it (C.D.); gulsumxyildirim@gmail.com (G.Y.); 2Faculty of Agricultural, Environmental and Food Sciences, Free University of Bozen-Bolzano, 39100 Bolzano, Italy; 3Department of Physics, University of Trento, 38123 Trento, Italy

**Keywords:** *Drosophila suzukii*, olfaction, antennal lobe, odour code, calcium imaging, preference assay

## Abstract

*Drosophila suzukii* is a major pest that damages berries and stone fruits by laying its eggs in ripening fruit still on the plant, in contrast to its sister species *Drosophila melanogaster*, which lays its eggs on overripe, fermenting fruit on the ground. Both species rely on their sense of smell to find such fruit, but how their brains process these odours and subsequently decide on different hosts is not yet fully understood. In this study, we hypothesised that the differences in behaviour may begin in the antennal lobes, the first brain regions of the olfactory system. Using advanced brain imaging techniques, we investigated how the two species respond to odours of ripe fruit, fermented fruit, leaves, and bacteria. We found structural differences in the antennal lobes and differences in the way the odours are represented in these areas of the brain, while behavioural experiments looking for direct differences in the attractiveness of the tested odours revealed no significant variation between the species. The differences in odour processing could form the basis for alternative species-specific pest control strategies that could reduce dependence on insecticides.

## 1. Introduction

The spotted wing Drosophila, *Drosophila suzukii* (Matsumura) severely damages the production of berry and stone fruits including strawberries, raspberries, blueberries, and cherries on most continents, leading to serious economic damages [1]. The pest is mostly managed by insecticides with concerning ecotoxicology profiles, some of which are authorized in organic farming [2,3,4]. Research is ongoing to find alternative tools, particularly to manipulate the flies’ behaviour [5,6,7]. Flies locate hosts and mates by sensing odorant volatile compounds using a highly specialized olfactory system. Using this chemosensory information is proving very valuable to help monitor and trap *D. suzukii* through attract-and-kill and push-pull methods, as well as lures [8,9]. An improvement in species specificity is, however, necessary, as most catches also include several other Drosophila species [10]. *D. suzukii* has the particularity of preferring ripening fruits in the plant canopy, in which they are physically able to oviposit, unlike the other co-existing Drosophila such as *Drosophila melanogaster*, which feed and reproduce on fermenting fruits on the ground [11,12,13,14,15]. The question is, how are the flies able to recognise suitable fruits before landing on them? The two different ecological niches of *D. suzukii* and *D. melanogaster* contain fruits of different maturity stages and different parts such as soil or canopy leaves and associated microbial and fungi communities. All these release numerous volatile organic compounds (VOCs) with overlapping and distinct properties, which are detected and used as cues by the flies [16]. However, yeast and bacterial communities are also associated with ripening fruits and play a significant role in enhancing the attractivity of ripening fruits in *D. suzukii* [17,18]. Thus, the distinction between *D. suzukii* and *D. melanogaster* niches is not so straightforward, and they are located very close to each other.

Airborne VOCs form odour plumes in varying ratios and quantities that are simultaneously or sequentially detected by the flies [19,20]. These VOCs activate different types of olfactory receptors (ORs) located on different types of olfactory receptor neurons (ORNs) on the antennae and maxillary palps [21]. The activation induces a series of action potentials that are relayed to the 58 glomeruli of the antennal lobes (Als), where ORNs expressing the same OR project their axons into the same glomerulus [22,23]. In the AL, input signals are processed by inter-glomerular coupling via lateral interneurons [24,25,26]. The output signal is then relayed via projection neurons to the mushroom bodies and lateral horns, where advanced processing takes place, including memory formation and behaviour [24,27,28,29]. The attractiveness of complex odours is likely evaluated in different steps along the processing pathway [30,31].

The Drosophila clade is an excellent model for studying chemosensory adaptation in closely related species with distinct ecological niches [32]. In *D. suzukii*, many genetic changes in the repertoire of olfactory genes suggest that the shift from fermenting fruits to ripening fruits in the suzukii lineage may be associated with functional changes in the detection of ripe and fermentation fruit odours by ORs [33,34,35]. Some anatomical changes in the peripheral olfactory system have already been reported, indicating a greater sensitivity to ripening volatiles: *D. suzukii* has twice as many ab2A and ab2B ORNs in the antenna compared to *D. melanogaster* [36] and half as many ab3A and ab3B [35]. However, at the receptor level, only few odours were found to be detected differently by the two species, and 32 out of 37 ORNs show conserved odorant binding affinities in *D. suzukii*, indicating that the species would not differ dramatically regarding the detection of environmental volatiles [35,37]. However, many fermenting products from fungal and microbial activity induce more attraction in *D. melanogaster* than in *D. suzukii* [38]. These fermenting cues are also seemingly driving a large part of the attraction to fruits in *D. suzukii* [18]. It is still unclear how odour information, although apparently detected in similar ways, is processed in the brain to trigger different behaviours in the two species, directing them to different environments. This lack of knowledge about specific attractants of *D. suzukii* limits the development of more efficient pest management tools. With a deeper understanding of the unique ecosystem perception of each species, we may be able to develop more species-specific tools.

Our objective was to explore how the first odour-processing step in the antennal lobe diverges between species: To this end, we used the latest genetic tools for functional neuroimaging available in *D. suzukii* [39], the genetically encoded calcium indicator GCaMP7s [40]. Using the UAS-GAL4 system, GCaMP can be co-expressed with Or83b (so-called Orco) in 80% of ORNs [41,42,43], including the ones affected by changes between lineages as mentioned above. Using two-photon in vivo calcium imaging, we investigated how the odour-activation pattern in the antennal lobe varies depending on odour type and species. We selected eight odours for which the detection was found to be particularly diverging between the two species due to genetic or behavioural evidence (Figure 1): (1) volatiles associated with ripening fruits: ethyl acetate and isoamyl acetate; (2) the leaf volatiles hexanal and (Z)-3-hexenyl acetate; (3) volatiles associated with fermenting fruits: 2-heptanone and 1-hexanol; and (4) the microbial volatiles acetic acid and geosmin [33,36,44,45,46,47]. We also hypothesized that odour mixtures may, due to species-specific changes in the odour interactions within the antennal lobe, result in varying attractiveness of ripening fruit odours. We thus assessed how such mixtures were encoded in the two species and how the two species responded behaviourally to these odour mixtures.

## 2. Materials and Methods

### 2.1. Insects

*Drosophila suzukii* transgenic lines *Orco-Gal4* [14] and *UAS-GCaMP7s-T2A-Tomato 3xp3RFP* [39] were kindly provided by Benjamin Prud’homme (IBDM, Marseille, France). *Drosophila melanogaster* transgenic lines *Orco-Gal4* (BDSC #26818) and *UAS-GCaMP7s* (BDSC #79032) were obtained from the Bloomington Drosophila Stock Center (University of Indiana, Bloomington, IN, USA). Wild-type flies were kindly provided by Gianfranco Anfora (C3A, University of Trento, San Michele all’Adige, Italy). Both species were maintained under controlled conditions at 22 ± 2 °C, 50–70% relative humidity, and a 16:8 light-dark cycle. Flies were reared on a medium containing water (1 L), yeast (15 mg), agar-agar (5 mg), soy flour (6 mg), corn flour (39 mg), malt syrup (23 mg), date syrup (13 mg), nipagin 50% in ethanol (5 mL), and propionic acid (3 mL). For *D. suzukii*, pieces of fresh fruits (blueberry, raspberry) were regularly added.

Six- to ten-day-old mated females *D. suzukii Orco-Gal4*, *UAS-GCaMP7s-T2A-tdTomato* and *D. melanogaster Orco-Gal4*, *UAS-GCaMP7s* were used for imaging experiments. Mated wild-type females were deprived of medium for four hours prior to the behavioural experiments to stimulate dispersal [48]. For the imaging experiments, they were not food-deprived to maximise their resilience during the preparation and imaging session. In all experiments, both fly species were prepared according to the same protocol to eliminate biases, e.g., due to different starvation effects.

### 2.2. In Vivo Imaging

To image odour-evoked responses in the Drosophila antennal lobe, flies were mounted and dissected following Silbering et al. [49] and Zanon et al. [50]. Time series were recorded using a two-photon microscope (Ultima IV, Bruker, Madison, WI, USA) with a Ti:Sa laser (Mai Tai Deep See HP, Spectra-Physics, Milpitas, CA, USA) tuned to 940 nm for GCaMP excitation. Images were acquired with a water-immersion objective (20 × NA 1.0, Olympus, Tokyo, Japan) at about 10 mW laser power to balance the signal-to-noise ratio against photo-damage effects. Recordings were performed with a frame rate of 10 Hz in the antennal lobe at three different depths with a vertical distance of approximately 20 µm to capture most glomeruli. Nineteen *D. suzukii* and 24 *D. melanogaster* females were imaged, with at least 10 responses collected per glomerulus.

Glomeruli were morphologically identified by acquiring z-stacks of the antennal lobe capturing both GCaMP and tdTomato signals. Localisation and identification of glomeruli were performed using FIJI (ImageJ 1.54f). The resulting masks were used to analyse the glomerular response time series.

### 2.3. Odour System Delivery

Odour stimuli were delivered by an eight-arm olfactometer with carbon-filtered humidified airflow containing odours in glass vials diluted by 1/200 *v*/*v* in paraffin oil. Single channels were operated by solenoid valves (LHDA0531115, The Lee Company, Westbrook, CT, USA) connected via a PCIe-6321 multifunction board (National Instruments, Austin, TX, USA) to a computer and controlled via a custom MATLAB script (R2019b, MathWorks, Natick, MA, USA). Constant airflow was maintained by switching between odour-filled and empty vials (containing 1 mL paraffin oil) [51]. Odour stimulation was fully automated and synchronised with microscopy, presenting 10 odour stimuli (pure odours or mixtures) 10 times to each fly. Stimuli lasted 3 s with a 10 s inter-stimulus interval and were quickly removed from the experimental area via an exhaust system.

Odorants of the highest purity available were purchased from Merck (Darmstadt, Germany): ethyl acetate (CAS 141-78-6), (Z)-3-hexenyl acetate (CAS 3681-71-8), 2-heptanone (CAS 110-43-0), Isoamyl acetate (CAS 123-92-2), 1-hexanol (CAS 111-27-3), acetic acid (CAS 64-19-7), hexanal (CAS 66-25-1), geosmin (CAS 19700-21-1), and paraffin oil (CAS 8042-47-5).

### 2.4. Identification of Glomeruli

To image the entire antennal lobe, brains were first dissected and stained, as in [52]. A reference atlas for glomerular identification and volume analysis was created for both species from volume images of the immunostained samples after 3D reconstruction and image segmentation using AMIRA 5.4 (Thermo Fisher Scientific, Waltham, MA, USA). Thirty-two glomeruli were identified based on published in vivo atlases [23,53]. Glomerular volumes were measured in 10 flies per species. Three functional imaging planes were selected that include landmark glomeruli easily identifiable via shape, position, and known response profiles to specific odours in *D. melanogaster*. Further glomeruli were identified based on their position with respect to the landmark glomeruli. Glomeruli in *D. suzukii* were identified based on the position of their equivalent in *D. melanogaster* and by a previously published atlas [54]. Only glomeruli clearly identified in at least 10 flies were considered.

### 2.5. Functional Imaging Data Post-Processing and Analysis

Functional data were analysed via custom MATLAB scripts. The relative fluorescence change was calculated by normalising the raw fluorescence signal *F*(*t*) by the average pre-stimulus signal $F_{b}$.$$\Delta F/F(t)=(F(t)-F_{b})/F_{b}$$

Average response amplitudes were computed by averaging Δ*F*/*F* over the 10 trials. Signals were further normalised across species so that the mean response amplitude averaged over all stimuli and all individuals was equal in both species.

The data normality was assessed using a Kolmogorov–Smirnov test. The dependence of the average response amplitudes on odour, species, glomerulus, and brain sides was statistically analysed using multiple-factor analysis of variance (ANOVA) and subsequent multiple comparison analysis with false discovery rate (FDR) correction. Left and right brain side glomeruli were pooled for subsequent analysis, as no significant differences were found (Appendix A), in contrast to other insect species [55].

Multidimensional odour response curves were visualised using principal component analysis (PCA). This allowed the 33-dimensional glomerular coding space to be reduced to three dimensions. The same transformation was applied to the data of both species so that the deviations between the temporal response curves in principal components visualise the dynamic differences in odour coding.

Euclidean distances were used to quantify between-species and within-species differences in the multi-dimensional response patterns. A hierarchical clustering analysis was performed on these multi-dimensional responses using Ward’s minimum variance method with the Euclidian distance as a metric. Odour responses are visualised as unique clusters if their distance is less than 70% of the maximum distance between all elements.

### 2.6. Four-Choice Arena Behavioural Assay

Behavioural responses to odours were assessed using a custom-made four-choice arena assay, testing the preference between four sources simultaneously presented in a static air environment. Rearing cages measuring 30 × 30 × 30 cm (BugDorm-1, MegaView Science, Taichung, Taiwan) were fitted with 30 × 30 × 0.01 cm platforms pierced with a Ø = 4 cm hole on each corner. A polyethylene square d-bottom flask (Flystuff 32-131F, Genesee Scientific, El Cajon, CA, USA) was attached under each hole via a 3D-printed funnel (tapering from 4 to 1 cm over 3 cm depth) as a trapping entrance. Each flask contained a water-soaked cotton ball and a Ø = 15 mm polyethylene container (BioScientifica, Rome, Italy) for the odours. Cages contained two flasks with containers filled with odours diluted 1/200 (*v*/*v*) in 1 mL paraffin oil (baits 1 and bait 2) and two control flasks: one with 1 mL pure paraffin oil and one empty. Seven conditions were created to test bait 1—ethyl acetate, isoamyl acetate, or acetic acid—against bait 2—the same odour as bait 1 or a mixture of the odour with acetic acid. Each bait combination experiment was replicated 8 times under changing bait positions. For each replication, 20–50 flies were briefly anaesthetised on ice and placed at the centre of the arena. After 24 h, the flies in each flask and the flies remaining on the platform were counted. Over 200 flies were tested per condition and species. The data normality was assessed using a Kolmogorov–Smirnov test. The proportions of flies for each bait were analysed using multiple-factor analysis of variance (ANOVA) and subsequent multiple comparison analysis with false discovery rate (FDR) correction.

## 3. Results

### 3.1. Identification and Volume Measurements of Glomeruli

In both species, the same 32 glomeruli were located in each antennal lobe (Figure 2a). They were most distinctly visible in three planes of the antennal lobe at different depths (Figure 2b). Near the top of the antennal lobe, directly under the head capsule, the following glomeruli were accessible: DA1, DA2, DA3, DA4l, DA4m, DL4, and D. About 20 µm deeper, the glomeruli DL1, DL3, DL5, DC1, DC2, DM2, DM3, DM5, DM6, VA1v, VA1d, VA6, and VM5v were reached. A further 20 µm deeper, the glomeruli DM1, DM4, VM2, VM5d, VM7v, VM7d, VA2, VA5, VA7, VC1, VC2, and VC3 became accessible. Some glomeruli were visible in multiple planes. The positions of these glomeruli were found to be similar in both species and consistent across the 10 flies analysed for each species.

The average overall volume of the antennal lobes in *D. suzukii* was 50% larger than in *D. melanogaster*. Therefore, structural differences between species for individual glomeruli were analysed based on their relative volume with respect to the total AL volume. A two-way ANOVA showed that this normalisation removed a general species dependence of the volume distribution (Appendix A). However, the species-glomerulus type interaction is highly significant, and a multiple comparison analysis showed significant differences in 6 out of the 32 analysed glomeruli: DL1, DL5, and DM4 were larger in *D. suzukii*, and VA1d, VA1v, VA2, and VA6 were smaller in *D. suzukii* compared to *D. melanogaster* (Figure 2c, Appendix A).

### 3.2. Response to Single Odours

Changes in fluorescence in response to odours were measured during and after a 3 s stimulus. The eight odours evoked specific response patterns in both species, which mostly rise and fall rapidly (Appendix A). The response patterns are similar but show selective differences between the species (Figure 3). To first investigate static response parameters, the average activity amplitude during the stimulus period was analysed via a four-way ANOVA (Appendix A), which gave no significant main effect of species, but the interaction effects between species and odour (*F*(7,15199) = 3.68, *p* = 5.6 × 10^−4^) and species and glomerulus (*F*(32,15199) = 5.81, *p* = 1.9 × 10^−23^) were highly significant. A multiple comparison analysis of the response amplitude averaged over all odours for the individual glomeruli gave significant differences in DL5, DM1, DM2, and VM2, which responded stronger in *D. melanogaster*, and in DM4 and VM7v, which responded stronger in *D. suzukii* (Figure 3a, Appendix A). A multiple comparison analysis of the response amplitude to individual odours averaged over all glomeruli gave a significant difference only for hexanal, for which the response in *D. melanogaster* was generally stronger (*p* = 0.001) (Figure 3b, Appendix A).

To highlight differences in the response dynamics, a PCA was performed, allowing visualisation of the subject-averaged response of all 33 glomeruli in three principal components (PCs) (Figure 3c). These first three PCs explained 77%, 9%, and 6% of the response pattern variance in both species, respectively. All odour response curves show relatively similar dynamics: a fast rise and a slightly slower decrease, which does not completely drop to the background activity level during the stimulus period. As for the differences between species, for some odours, curves change their relative position to others; this is most evident for isoamyl acetate. A general difference is that PC1 has a smaller amplitude in all odour responses of *D. suzukii.*

To quantify the differences between these general odour codes, Euclidean distances (ED) were measured between these multi-dimensional odour response vectors, time-averaged during the stimuli. To distinguish species-specific differences from simple variations across subjects, EDs between flies of different species were compared with the EDs between fly pairs within each species (Appendix A). While the variation and thus the ED within the *D. melanogaster* group was as high (or higher) as the difference between species, in the *D. suzukii* group, the odour code seems to be preserved much stronger so that the ED within this group was significantly smaller than the ED between species for all odours.

Next, a hierarchical cluster analysis (HCA) was used to sort odour codes according to their similarity measured by the ED. The odours clustered in two major groups, one containing most leaf and bacterial odours and the second containing ripening odours (Figure 3d). Fermenting odours were present in both groups. The two species differed by the clustering of hexanal, which was closer to ripening fruit odours in *D. melanogaster*, while it clustered with leaf and bacterial odours in *D. suzukii*. 2-heptanone and isoamyl acetate were more closely clustered in *D. suzukii* (*ED* = 6.4) compared to *D. melanogaster* (*ED* = 8.9). In addition, acetic acid was closer to 1-hexanol, geosmin, and (Z)-3-hexenyl acetate in *D. suzukii* (*ED* = 5.1) than in *D. melanogaster* (*ED* = 8.2).

### 3.3. Responses to Odour Mixtures

In the next experiment, two bacterial odours, acetic acid and geosmin, were added to the ripening fruit odours ethyl acetate and isoamyl acetate to test the extent to which this mixture alters the response patterns to the pure components in both species (Appendix A and Figure 4).

A four-way ANOVA (Appendix A) gave, as for pure odours, no main effect of species but strong interaction effects between species and odour (*F*(7,15199) = 11, *p* = 1.3 × 10^−13^) and species and glomerulus (*F*(32,15199) = 6.1, *p* = 2.3 × 10^−25^). The multiple comparison analysis of the response amplitude for individual glomeruli averaged over all odours showed significant differences in, again, 6 out of the 32 glomeruli. DL5 and DM2 responded stronger in *D. melanogaster*, and DM4, DM5, DM6, and VM5v responded stronger in *D. suzukii* (Figure 4a, Appendix A). The multiple comparison analysis of the response amplitude for individual odours averaged over all glomeruli gave two highly significant differences; both are mixtures: ethyl acetate + geosmin elicits stronger responses in *D. melanogaster* (*p* = 3.9 × 10^−5^) and isoamyl acetate + acetic acid in *D. suzukii* (*p* = 1.1 × 10^−11^) (Figure 4b, Appendix A).

The response dynamics analysis via PCA allowed for the explanation of 88%, 5%, and 2% of the response patterns in both species with the first three components, respectively. Plotting the subject-averaged temporal response curves for the individual odours (Figure 3c) again shows less contribution of the first PC1 to the response in *D. suzukii* and some changes in positions of the response curves between species. Noteworthy are ethyl acetate and its mixtures with acetic acid and geosmin, which show notable shifts in odour space between the species. Quantifying the distances in multi-dimensional coding space, the same approach as for the pure odours shows that for *D. melanogaster*, the within-species variance is as high or higher than the variance between species. In *D. suzukii*, the odour code seems again better conserved and thus significantly different from the Euclidean distance between species, but now with two exceptions, both of them mixtures: ethyl acetate + acetic acid and isoamyl acetate + geosmin (Appendix A).

The HCA showed that the odours and mixtures clustered diversely in both species. In *D. melanogaster*, two fundamental clusters could be identified. The mixtures with acetic acid grouped with isoamyl acetate and geosmin. The mixtures containing geosmin instead grouped with ethyl acetate. However, in *D. suzukii*, three clusters were identified. As for *D. melanogaster*, the mixtures containing geosmin clustered, but interestingly also with isoamyl acetate and the mixture of ethyl acetate + acetic acid. Another cluster was formed by ethyl acetate and the isoamyl acetate + acetic acid mixture. The bacterial odours acetic acid and geosmin grouped separately from fruit odours and mixtures (Figure 4d).

### 3.4. Behavioural Responses to Mixtures and Components

The behavioural responses of female *D. suzukii* and *D. melanogaster* were assessed in a four-choice arena assay (Figure 5a). The choice was given between a single ripe fruit odour (bait 1) and a mixture of the single odour and acetic acid (bait 2). Each set contained two controls, paraffin oil (control 1) and an empty vial (control 2). In additional control experiments, bait 1 and 2 were either identical ripe fruit odours or both acetic acid.

Experiments were separately analysed for each of the three reference odours (bait 1, Figure 5b–d) via a two-way ANOVA with factors bait 2 and species (Appendix A). Species had no significant effect in any experiment; the interaction between bait 2 and species was significant only when bait 1 was ethyl acetate, and the effect is likely due to differential responses to the paraffin oil control. In a multiple comparison analysis with FDR correction, this effect also lost significance (Appendix A).

The main effects of odours were highly significant, but again, there were no species differences in preferences for individual odours (Figure 5b–d, Appendix A). In experiments with pure ethyl acetate asbait1, both fly species preferred the ethyl acetate + acetic acid mixture over pure ethyl acetate (*D. melanogaster*: *p* = 0.008, *D. suzukii*: *p* = 0.02) and pure acetic acid (*D. melanogaster*: *p* = 0.02, *D. suzukii*: *p* = 0.011). There was no significant preference between pure ethyl acetate and pure acetic acid (Figure 5c). In the experiments with isoamyl acetate, the mixture between isoamyl acetate + acetic acid was preferred over pure isoamyl acetate (*D. melanogaster*: *p* = 0.0067, *D. suzukii*: *p* = 0.004), again identically in both species (Figure 5d).

## 4. Discussion

Many fermentation products from fungi and microorganisms are more attractive to *D. melanogaster* than *D. suzukii* [38]. However, these cues drive much of *D. suzukii’s* attraction to fruits [18,56]. We investigated differences in antennal lobe coding of ripe, overripe, leaf, and microbial odours between species.

### 4.1. Glomerular Size Differences Between Species May Reflect Varying ORN Numbers

To evaluate the structural dimension of this adaptation, we precisely measured the glomerular volumes. Even after normalising for *D. suzukii’s* larger brain, six glomeruli had significantly different volumes. Three of these were larger in *D. suzukii* (Figure 2c), including DM4, which is consistent with previous findings that *D. suzukii* has almost twice as many corresponding sensilla ab2, including ab2A, as *D. melanogaster* (22 vs. 12). Previous studies have shown correlations between glomerular size, ORN abundance, and ORN activity [23,35,54]. DM4 is tuned to ripening fruit odours, while DL5 and DL1, which we also found to be larger in *D. suzukii*, mostly respond to green leaf volatiles and repellent odours, respectively, in *D. melanogaster* [35,57]. In contrast, the four larger glomeruli in *D. melanogaster* (VA1d, VA1v, VA2, VA6) are tuned to fermenting odours [35,57]. Therefore, our results support the hypothesis that *D. suzukii* allocates more energy to the detection of ripening fruit and green leaves [33,36].

### 4.2. Averaged Glomerular Responses in the Two Species Were Similar, Yet the Odour Code Changed

We tested eight odours from leaves, ripening fruit, and fermenting fruit-associated microorganisms, imaging glomerular responses in both species.

#### 4.2.1. Single Glomerular Response Changes Connect to ORNS Changing Their Odour Specificity

The average glomerular response amplitude to all odours was found to be significantly different in five glomeruli (Figure 2). DM4 and VM7 showed stronger responses in *D. suzukii*, which aligns with their tuning to ripening fruit odours in *D. suzukii* [33,58]. DM2 responded significantly more strongly in *D. melanogaster*, consistent with the discovery that the associated sensilla ab2B and ab3A are selective to yeast-specific vital odours [59] but have shifted to detect ripening fruit odours, such as isoamyl acetate, in *D. suzukii* [35]. The corresponding receptors, Or85a and Or22a, respectively, were considered to be completely replaced [33,34]. The ab3A has a clear role in host shift due to a high variability of the associated receptor Or22a across Drosophila species including *D. suzukii* [35,60,61,62].

DL5 was found to be less strongly activated in *D. suzukii* despite its association with deterrent valence [58], suggesting that it became less deterrent in *D. suzukii*, as the associated receptor Or7a was not as strongly activated in *D. melanogaster*. In our recordings, the leaf volatiles hexanal and (Z)-3-hexenyl acetate activated DL5.

The response to hexanal stood out as the only odour with significantly different responses between species across all glomeruli, with *D. suzukii* showing a lower average response (Figure 3b). The overall odour code of hexanal clustered more with bacterial and fermenting odours in *D. suzukii* while with ripening fruits in *D. melanogaster* (Figure 3d). Yet, hexanal has previously been found to be attractive for *D. suzukii* [46]. One hypothesis is that response changes in glomeruli such as DL5 that encode valence cause this reversal in attraction, drawing *D. suzukii* to canopies and fruits despite hexanal-emitting leaves. Similar observations were made in studies identifying a prevalent role of beta-cyclocitral, a bacterial compound associated with canopy leaves in strawberry plants [36]. The strong deviation in the odour code for hexanal could suggest that leaf odours may actually play a larger role in host selection than suspected. This should be investigated in future behavioural studies. The overall responses of DM1 and VM2 were also stronger in *D. melanogaster* linked to ripening and fermenting odours, respectively [33,58] but were not found to be divergent between species in previous studies.

#### 4.2.2. Odors Produce Different Glomerular Patterns in the Antennal Lobes of Both Species

Aside from hexanal, no odours showed significantly different response amplitudes if averaged across glomeruli. Thus, we investigated the multidimensional glomerular code in both species, including dynamical features. We observed very heterogenic response dynamics across glomeruli, ranging from phasic responses only at the odour onset, via responses that well resembled the full odour pulse shape, to tonic responses well beyond the stimulus duration. The post-stimulus phase was therefore included in the overall activation pattern analysis to capture the whole complexity [63].

The subject-averaged temporal response curves show clear species-specific differences when plotted in common principal components (Figure 3).

Why this is not apparent from the statistical analysis of the average response amplitudes became clear when Euclidean distances between the odour codes were analysed in single fly pairs (Appendix A): In *D. suzukii*, the response variance across subjects was small, resulting in a consistent shift from *D. melanogaster.* In contrast, a high within-species variance in *D. melanogaster* obscured this shift.

#### 4.2.3. Odour Codes Clustered Differently in the Two Species

The difference between the species can be seen above all in the different relative positions of the odour response curves in the coding spaces. This means that the similarity between odours is perceived differently in the two species. This is also reflected in the clustering of the odour codes. In both species, the fermenting, leaf, and bacterial odours grouped, while ripening fruits clustered separately (Figure 3). The major difference was in the clustering of hexanal, which, as discussed above, had the most divergent odour code between the species. Two other compounds clustered differently and deserve attention: Firstly, isoamyl acetate was closer to overripe fruit odours in *D. suzukii*. This is interesting, in that a previous study found that not only did the expression of olfactory receptors tuned to isoamyl acetate increase in *D. suzukii* [45], the behavioural responses to this compound were also altered: it is found in small amounts in ripening fruits and is attractive to *D. suzukii* [46,64], and it is found in larger amounts in fermenting fruits and associated yeasts, where it is less attractive [64,65]. Secondly, acetic acid clustered closer to other microbial odours in *D. suzukii* than in *D. melanogaster*. Acetic acid is found in many bacterial and fungal species associated with Drosophila, and it is attractive to both species [47,65,66].

Regarding the distinction between ripening and fermenting fruit odours and bacterial odours, we found a good separation of the multi-glomerular odour codes in both species but no significant differences in the glomerulus-averaged response amplitudes. This highlights the complexity of the odour code, which goes beyond static amplitude coding [52].

#### 4.2.4. Geosmin Was Not Found to Be Coded in a Labelled Line

We observed species-specific differences in geosmin’s glomerular activation pattern, but no significant difference in the average response amplitude across all glomeruli. Geosmin, a bacterial compound aversive to adult *D. suzukii*, has previously been found to activate primarily the receptor Or56a, projecting to glomerulus DA2 in both species [37,44]. However, we observe a broader response spectrum without DA2 involvement. One possible explanation is the larger concentration with respect to previous experiments. Geosmin was found to be detected in Drosophila already at a concentration of 10^−8^, saturating the ORNs at 10^−4^ [44]. In our experiments, all odours were administered at a 0.5 × 10^−2^ dilution. Glomerular response patterns are known to become broader for higher concentrations, and a recent work in bees has shown both experimentally and in simulations that projection neuron responses can, beyond saturation, even drop to zero due to inhibitory coupling between glomeruli [67].

### 4.3. Mixture Response Patterns Reveal Synergistic and Suppressive Glomerular Interactions, Though Species Differences Did Not Reflect in Behaviour

Ripe fruits also host yeast species whose volatiles are often associated with fermenting fruits [18,45,59], which are detected by *D. suzukii* and contribute to the attraction to ripe fruits [8,68]. Therefore, *D. suzukii* flies must detect and process mixtures of these volatiles with odours of host plants. To test species differences, we added bacterial odours, acetic acid and geosmin, to fruit odours, ethyl acetate and isoamyl acetate, and analysed the mixture response patterns.

The ethyl acetate + geosmin mixture triggered a significantly higher overall response in *D. melanogaster* compared to *D. suzukii*, while the isoamyl acetate + acetic acid mixture produced significantly stronger responses in *D. suzukii* (Figure 3). The multi-glomerular coding patterns were shifted between species. *D. suzukii* showed again small within-species variance so that the between-species shift was found to be significant for six of the eight odours (Appendix A). In contrast, *D. melanogaster* showed again a within-species variance as high or higher than the between-species shift of the odour code.

Looking at the clustering of odour codes within species (Figure 3), especially the bacterial odours, acetic acid and geosmin clustered separately from fruit odours and mixtures in *D. suzukii.* The mixtures grouped differently depending on the species, reflecting distinct odour codes. Four glomeruli (DM4, DM5, DM6, and VM5v) showed increased activity in *D. suzukii* across all mixtures and their components, while two (DL5 and DM2) showed decreased activity with respect to *D. melanogaster* (Figure 3). In particular, DL5 was activated by a deterrent compound that was found to inhibit DM4 and DM2 in an earlier study on *D. melanogaster* [25].

These differences between components and mixtures likely result from glomerular coupling via lateral neurons in the antennal lobe [25,69], while different valences are imprinted by either the modified glomerular code [30] or higher processing centres [26].

In the behavioural experiments, *D. suzukii* and *D. melanogaster* showed similar responses. In particular, the odour mixtures containing combinations of fruit and microbial volatiles were preferred over their individual components, which is consistent with earlier works, where these mixtures led to increased catches [65,70,71,72]. In both species, acetic acid was as attractive as ethyl acetate and more attractive than isoamyl acetate. Previous work showed that acetic acid was equally attractive to both flies at similar doses (<0.5%) [47]. These results emphasise the role of microbial volatiles in influencing olfactory-driven behaviours and highlight the potential complexity of odour interactions in shaping species-specific ecological strategies.

Despite the differences in odour coding, *D. suzukii* and *D. melanogaster* showed no significantly different behaviour in responses to single odours and mixtures. We must conclude that these odours are not the components that trigger the species-specific ecological behaviours or that additional contextual factors are needed. Therefore, further studies are needed to identify such active odorants and odour mixtures.

## 5. Conclusions

This study shows how different olfactory processing might support ecological specialisation in these closely related species.

Morphological imaging revealed differences in the size of glomeruli in the antennal lobes of both species, with *D. suzukii* having larger glomeruli for odours associated with ripening fruits and green leaves. These structural differences likely reflect different ORN populations and are consistent with ecological preferences.

Functional imaging showed nuanced differences in the olfactory processing of ripe and overripe fruits, leaves, and microbial volatiles. These persistent differences reflect a general species-dependent shift in odour coding. Differences were particularly evident in mixtures. A consequence could be a different recognition sensitivity for odours of ripe and fermenting fruits.

Despite the differences in neuronal coding, both species showed similar behavioural responses to the tested odours and mixtures, suggesting that these substances in their pure form do not contribute to the deviating host plant-selection behaviour. This calls for further behavioural experiments to reveal the active components in this context.

A next valuable step in tracking differences in odour processing could be a comparative study of neuronal patterns in higher brain centres of both species, such as the lateral horn, which is expected to assess odour valence.

The key finding of this study, a species-specific shift in odour coding, motivates further research on pest control strategies based on species-specific attractive/repulsive odour components for *D. suzukii*, potentially reducing dependence on insecticides and improving ecological tolerance.

## Figures and Tables

**Figure 1 insects-16-00084-f001:**
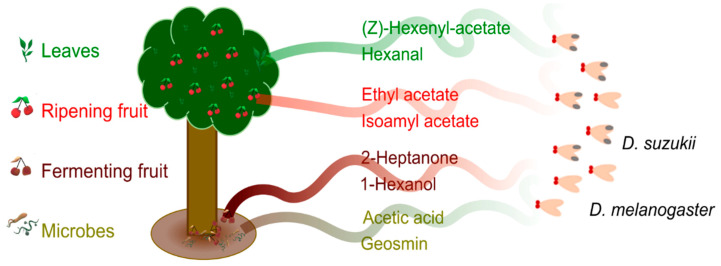
Eight volatiles were selected from the two ecological niches occupied by *Drosophila suzukii* and *Drosophila melanogaster*. The ripe fruit-associated odours include ripening fruit volatiles (ethyl acetate, isoamyl acetate) and canopy leaf volatiles (hexanal, (Z)-3-hexenyl acetate). The fermenting fruit-associated odours include overripe fruit volatiles (2-heptanone, 1-hexanol) and microbial volatiles (acetic acid and geosmin).

**Figure 2 insects-16-00084-f002:**
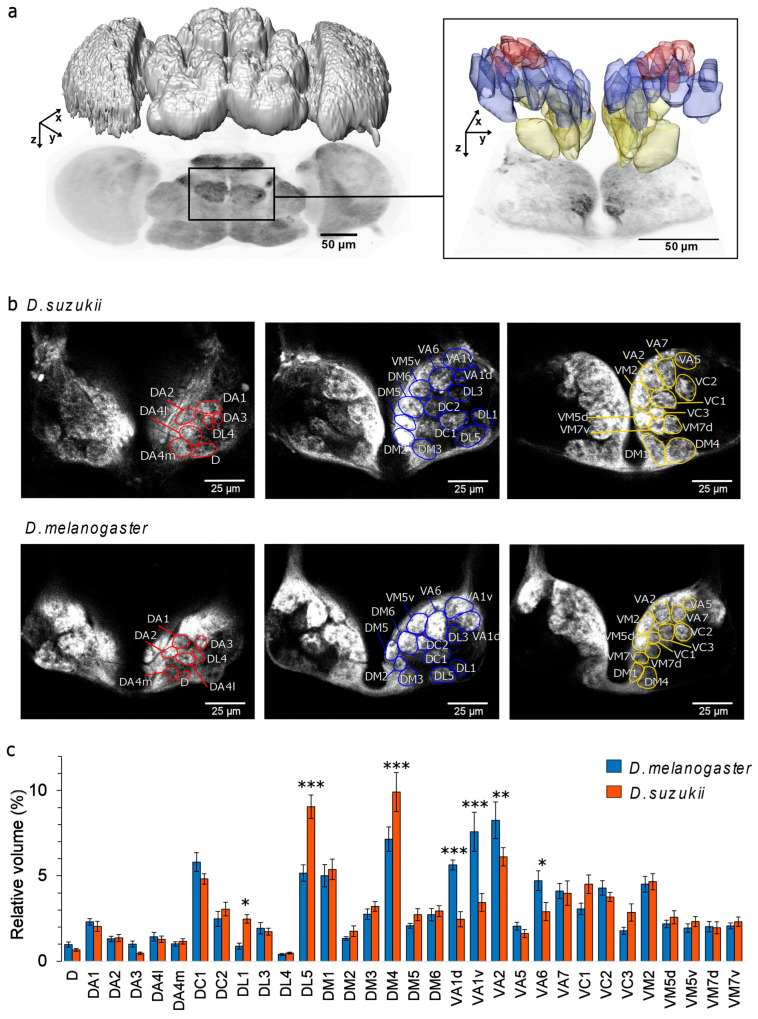
Characterisation of glomeruli in the antennal lobe of *D. suzukii* and *D. melanogaster.* (**a**) Image of the whole immunostained brain of *D. suzukii* using synapsin-dsRed antibodies (**left**) and imaging of the two antennal lobes with ORNs stained by tdTomato (**right**). On top of a projection view is the threshold-segmented volume image of the whole brain (**left**) and volume images of segmented single glomeruli in both antennal lobes (**right**). Arrows show image plane (x, y) and stacking direction (z). Glomerular colours correspond to the three imaging layers in (**b**). (**b**) Mapping of the identifiable glomeruli in three cross sections (with increasing depth) of the antennal lobes in *D. suzukii* (1-3), and *D. melanogaster* (1′-3′), corresponding to the focal planes used to record the neural activity. (**c**) Mean ± standard error of the mean (SEM) of the relative volume [%] of 32 identified glomeruli in the right antennal lobes of *D. melanogaster* (blue bars) and *D. suzukii* (orange bars), Significant statistical differences (multiple comparison analysis with FDR correction) between species are labelled according to their significance probability as * *p* < 0.05, ** *p* < 0.01, *** *p* < 0.001, *n* = 10 per species.

**Figure 3 insects-16-00084-f003:**
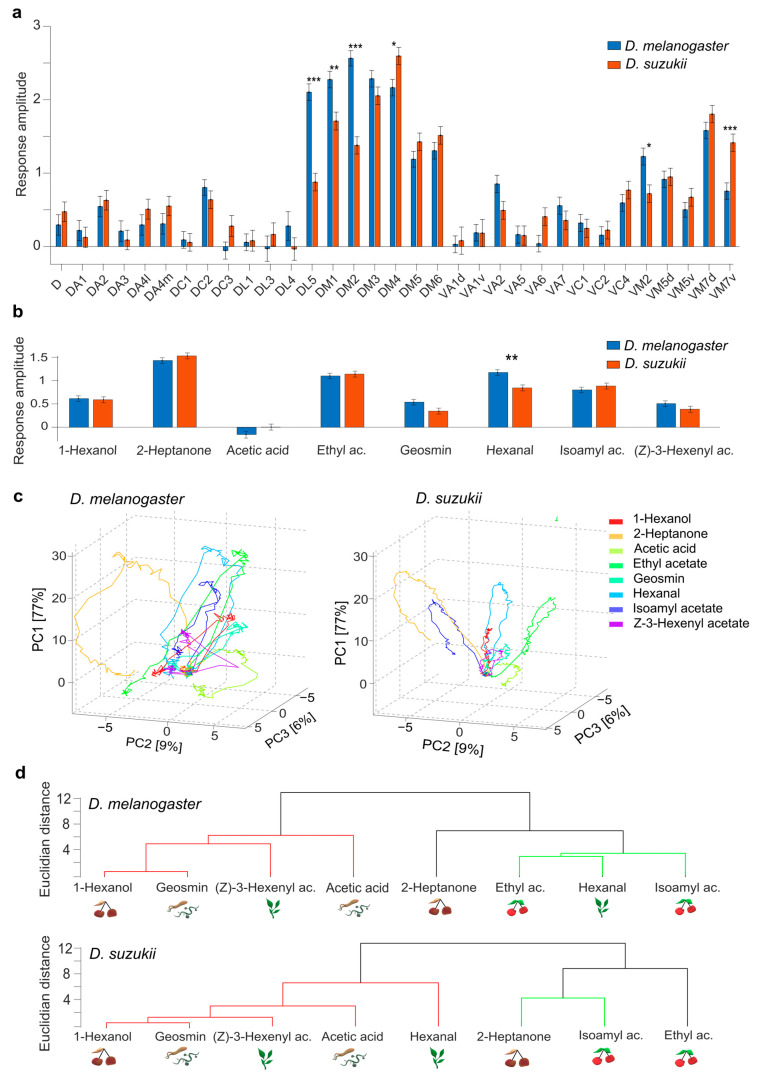
Odour response maps and dynamics for eight odorants in *D. suzukii* and *D. melanogaster.* (**a**) Mean ± SEM response amplitude in each glomerulus, averaged over all odours. (**b**) Mean ± SEM response amplitude to each odour, averaged over all glomeruli. *D. suzukii* is shown in orange, and *D. melanogaster* is shown in blue. Significant statistical differences (multiple comparison analysis with FDR correction) between species are labelled according to their significance probability as * *p* < 0.05, ** *p* < 0.01, *** *p* < 0.001. (**c**) Temporal response curve in a coding space reduced by PCA from 33 glomerular dimensions to 3. The curves show the signal increase and decrease during the stimulus period. The distance between the curves at the inflexion point is a measure of the discriminability of the odours. The coordinate system is identical in both plots, so the differences in the response curves between the left and right plots illustrate the difference in the odour code between the species. (**d**) Hierarchical cluster analysis (HCA) using Ward’s method with Euclidean distances (ED) between odour responses in *D. melanogaster* (upper) and *D. suzukii* (bottom); the *y*-axis quantifies the ED between clusters. The different colours mark clusters in which the ED is less than 70% of the maximum distance between all elements. Icons illustrate the ecosystem where the odorant is most abundant: fermenting fruits (rotten cherries), ripening fruits (ripe cherries), leaves on the canopy (green leaves), and microbes (bacteria and fungi).

**Figure 4 insects-16-00084-f004:**
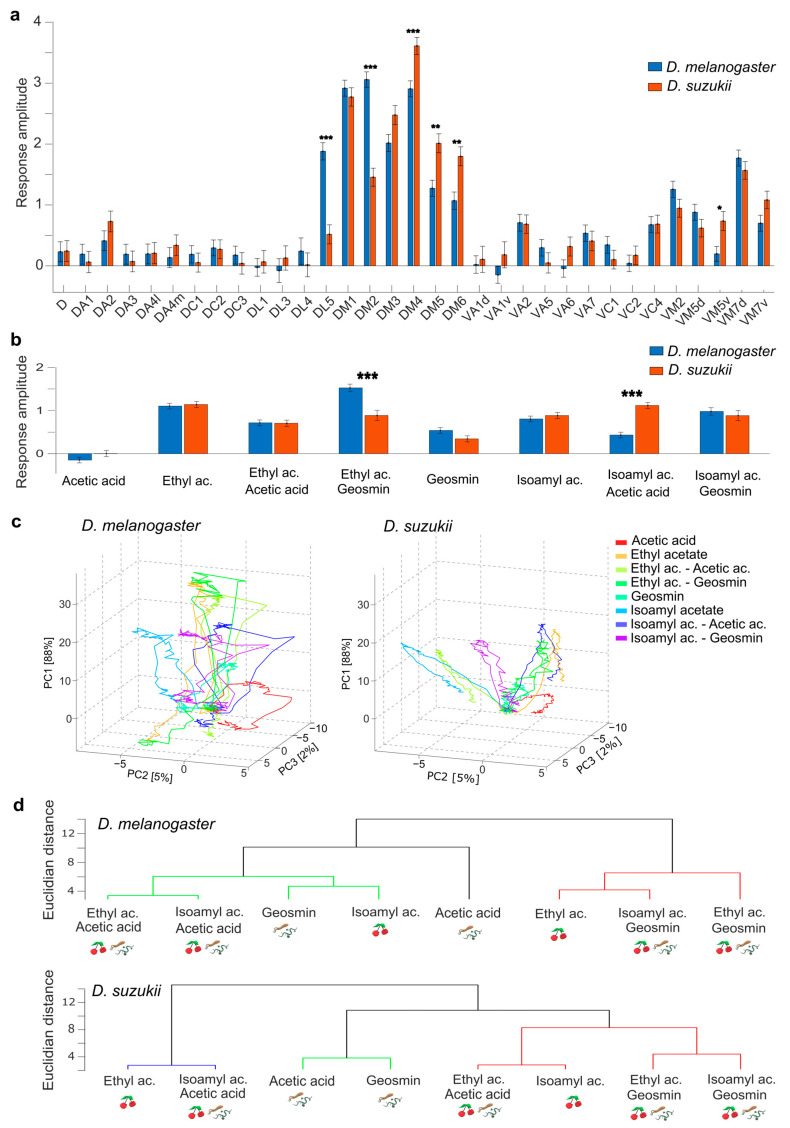
Odour response maps and dynamics for mixtures and their components in *D. suzukii* and *D. melanogaster.* (**a**) Mean ± SEM response amplitude in each glomerulus, averaged over all odours. (**b**) Mean ± SEM response amplitude to each odour, averaged over all glomeruli. *D. suzukii* is shown in orange, and *D. melanogaster* is shown in blue. Significant statistical differences (multiple comparison analysis with FDR correction) between species are labelled according to their significance probability as * *p* < 0.05, ** *p* < 0.01, *** *p* < 0.001. (**c**) Temporal response curve in a coding space reduced by PCA from 33 glomerular dimensions to 3. The curves show the signal increase and decrease during the stimulus period. The distance between the curves at the inflexion point is a measure of the discriminability of the odours. The coordinate system is identical in both plots, so the differences in the response curves between the left and right plots illustrate the differences in the odour code between the species. (**d**) Hierarchical cluster analysis (HCA) using Ward’s method with Euclidean distances (ED) between odour responses in *D. melanogaster* (upper) and *D. suzukii* (bottom); the *y*-axis quantifies the ED between clusters. The different colours mark clusters in which the ED is less than 70% of the maximum distance between all elements. Icons illustrate the ecosystem where the odorant is most abundant: ripening fruits (ripe cherries) and microbes (bacteria and fungi).

**Figure 5 insects-16-00084-f005:**
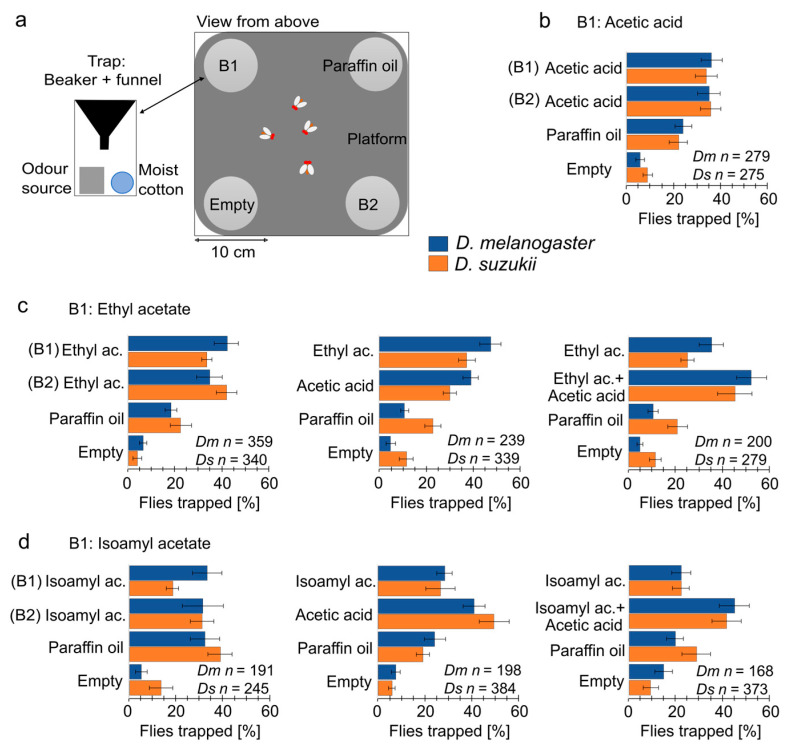
Four-choice arena assays with *D. suzukii* and *D. melanogaster*. (**a**) Four-choice cage assay to assess behavioural responses to four odours by groups of adult *D. suzukii* and *D. melanogaster*. Traps containing baits (B1 and B2) and controls (paraffin oil and empty) were positioned at each corner of the cage with a platform level with the entrance of the traps. (**b**–**d**) Mean ± SEM proportion of flies [%] counted in all four traps after 24 h in different choice combinations: Bait 1 (B1) contained either acetic acid (**b**), ethyl acetate (**c**), or isoamyl acetate (**d**). Bait 2 (B2) contained either the same odour as B1 or a mixture with acetic acid. Twenty to fifty flies were released in the middle of the platform, and each experiment was repeated eight times; the total fly count in all flasks across these repetitions is given in the plots.

## Data Availability

Detailed statistical data are provided in the Appendix A. Raw data from brain imaging and behavioural experiments are available from https://github.com/NeurophysicsTrento/Comparative-Study-Drosophila (accessed on 12 January 2025).

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
