# Peer review of "Differential Coding of Fruit, Leaf, and Microbial Odours in the Brains of *Drosophila suzukii* and *Drosophila melanogaster"

_insects, 2025, doi:10.3390/insects16010084_

Round 1

Reviewer 1 Report

Comments and Suggestions for Authors

Dumenil et al. investigate the difference in odor processing and odor preference between Drosophila melanogaster and Drosophila suzukii. They perform anatomical analysis and calcium imaging in the antennal lobe and behavioral experiments with a subset of ecologically relevant odorants. Odors were chosen based on the different lifestyles of the species – D. mel prefers to lay eggs in overripe fruits, whereas D. suzukii lays eggs in ripe fruits. The authors find differences in glomerular size and also in response patterns across glomeruli for odors and odor mixtures. However, the odors and odor mixtures did not elicit different behavioral responses in the different species.

The study uses a novel method, calcium imaging in the ORNs of Drosophila suzukii in response to different odors. Even though they can detect differences in the response patterns across glomeruli, these do not seem to have any relevance to the behavior of the fly. Thus, the knowledge advancement of the study is limited and I would suggest a major revision and additional behavior experiments.

Some comments and suggestions:

Several glomeruli respond differently across species and odor/odor mixtures; however, no consistent pattern arises regarding the ecological niche of the species. The authors describe that hexanal induces the biggest difference in response across all glomeruli between the two species. The authors might want to include this odor in the behavioral studies, as it might lead to behavioral differences, potentially also in a mixture with the other odors. Generally, more odor mixtures in different concentrations should have been tested for the behavior to exclude any differences.

Figure 5 shows behavioral data and is very hard to understand. For example, in Figure 5b data is shown for AA in bait 1 – is this the control experiment where AA was also in bait 2? Why are there not 4 bars shown – one for bait 1, bait 2, paraffin, and nothing? Also, in the last two figures – is the paraffin and nothing control pooled across experiments? How many flies had been trapped in bait 1 in each experiment? Did this differ when there was a choice between different odors in bait 2?

The authors food-deprived flies for 4 hours for behavior experiments – have the flies been treated the same way for calcium imaging experiments? Starvation could influence ORN activation patterns.

In Table S13 – the exact same values are shown for bait 1 Isoamyl acetate experiments across D. melanogaster and D. suzukii responses. Is this true?

Author Response

Comment 1: Dumenil et al. investigate the difference in odor processing and odor preference between Drosophila melanogaster and Drosophila suzukii. They perform anatomical analysis and calcium imaging in the antennal lobe and behavioral experiments with a subset of ecologically relevant odorants. Odors were chosen based on the different lifestyles of the species – D. mel prefers to lay eggs in overripe fruits, whereas D. suzukii lays eggs in ripe fruits. The authors find differences in glomerular size and also in response patterns across glomeruli for odors and odor mixtures. However, the odors and odor mixtures did not elicit different behavioral responses in the different species.

The study uses a novel method, calcium imaging in the ORNs of Drosophila suzukii in response to different odors. Even though they can detect differences in the response patterns across glomeruli, these do not seem to have any relevance to the behavior of the fly. Thus, the knowledge advancement of the study is limited and I would suggest a major revision and additional behavior experiments.

Response 1: We thank the reviewer for his feedback and all the constructive comments below. Regarding the general comment on the scientific value, we kindly ask the reviewer to consider the following question: This study represents the first quantification of differences in the structure and function of the peripheral olfactory system in these two Drosophila species. If we had submitted only the neuroimaging data, would he still have dismissed these results as providing limited advancement of knowledge? In our view, negative results also contribute valuable knowledge to science, which is why we decided to include also the behavioural study, even though it did not prove our hypothesis of a significant difference in behaviour. While we agree that further behavioural studies are necessary to build upon these results, we respectfully disagree with the assessment that this study lacks significant knowledge advancement.

Comment 2: Several glomeruli respond differently across species and odor/odor mixtures; however, no consistent pattern arises regarding the ecological niche of the species. The authors describe that hexanal induces the biggest difference in response across all glomeruli between the two species. The authors might want to include this odor in the behavioral studies, as it might lead to behavioral differences, potentially also in a mixture with the other odors. Generally, more odor mixtures in different concentrations should have been tested for the behavior to exclude any differences.

Response 2: The behavioural experiments conducted here aimed to explore the effect of mixtures of mature volatiles and bacterial odours on host selection. Given the different neuronal response patterns to hexanal, the role of leaf odours in host choice is an interesting question that should indeed be addressed in future studies. Unfortunately, it is not possible to collect such data within the revision of this article. We thank the reviewer for this suggestion which we added to the discussion in lines 439ff, 528ff, 546ff.

Comment 3: Figure 5 shows behavioral data and is very hard to understand. For example, in Figure 5b data is shown for AA in bait 1 – is this the control experiment where AA was also in bait 2? Why are there not 4 bars shown – one for bait 1, bait 2, paraffin, and nothing? Also, in the last two figures – is the paraffin and nothing control pooled across experiments? How many flies had been trapped in bait 1 in each experiment? Did this differ when there was a choice between different odors in bait 2?

Response 3: We agree that Figure 5 was confusing, we had tried to compress the results of 7 individual experiments into 3 graphs. We refrained from this approach and present now each experiment individually, leading to 4 data points per species per experiment.

We have also added the total number of flies collected in all flasks to each subplot of Fig. 5 so that the absolute fly number for each bait can be determined.

Finally, we tried to further clarify the description of the experiment in the Methods section, line 209ff, the Results section 356ff, and the caption of the new Fig. 5.

Comment 4: The authors food-deprived flies for 4 hours for behavior experiments – have the flies been treated the same way for calcium imaging experiments? Starvation could influence ORN activation patterns.

Response 4: Indeed food deprivation was shown to induce changes in detection of food sources and associated volatiles. Flies were food deprived for 4 hours for the behavioural experiment to increase the dispersal of the flies, especially for D. suzukii.  With 4 h starvation we reached an optimum dispersal.  We added a reference in the Method section, line 129ff to clarify our motives.

In the imaging experiments, flies were not starved before preparation to ensure they were of the best health before the dissection, which proved to be very delicate. The transgenic lines were indeed weaker compared to the wild types, so the priority was to have a live and healthy animal throughout the imaging experiment (1 - 1.5 h of recording). However, we ensured that both fly species were treated exactly the same so that differences in the odour response profiles could not be due to different levels of starvation. We added also this information to the Methods section, line 129ff.

Comment 5: In Table S13 – the exact same values are shown for bait 1 Isoamyl acetate experiments across D. melanogaster and D. suzukii responses. Is this true?

Response 5: We would like to thank the reviewer for pointing this out. This was an error in the compilation of the table which we have now corrected.

Reviewer 2 Report

Comments and Suggestions for Authors

The manuscription “Differential coding of fruit, leaf and microbial odours in the 2 brains of Drosophila suzukii and Drosophila melanogaster” compares 3 aspects at the antennal lobes in the olfactory systems of D. suzukii and D. melanogaster: 1) glomeruli volumes, 2) Calcium imaging responses toward behaviorally important olfactory cues in the identified glomeruli of these flies and 3) behavior responses of the flies toward the chemicals. The main findings are there were differences in glomeruli volumes between the species and detected different responses in some identified glomeruli which are expected. However, the behaviors responses are somewhat unexpected or less clear.  Overall, the manuscript is clearly written and well organized. It is clear what have been done and what questions the experiments attempt to address. However, the result section is extremely hard to read and, in particular, is difficult to make sense out of the (complex) analysis. The vast amount of supplement data did not help much neither. I have a hard time to understand the PC (principal components) analysis illustrations (Figure 3 and 4 panels C). Some additional explanations in either the legends or the text would help. There is not a clear conclusion I can learn from the paper which is really frustrating as the methodology seems to be sound and huge quantity of data was analyzed extensively (maybe overly tortured?).

Minor:

Line 133: “19 D. suzukii…” should be “Nineteen D. suzukii…”

Line184: “variance method with the Euclidian Euclidean distance as a metric...."

Author Response

Comment 1:  The manuscription “Differential coding of fruit, leaf and microbial odours in the 2 brains of Drosophila suzukii and Drosophila melanogaster” compares 3 aspects at the antennal lobes in the olfactory systems of D. suzukii and D. melanogaster: 1) glomeruli volumes, 2) Calcium imaging responses toward behaviorally important olfactory cues in the identified glomeruli of these flies and 3) behavior responses of the flies toward the chemicals. The main findings are there were differences in glomeruli volumes between the species and detected different responses in some identified glomeruli which are expected. However, the behaviors responses are somewhat unexpected or less clear.  Overall, the manuscript is clearly written and well organized. It is clear what have been done and what questions the experiments attempt to address. However, the result section is extremely hard to read and, in particular, is difficult to make sense out of the (complex) analysis. The vast amount of supplement data did not help much neither. I have a hard time to understand the PC (principal components) analysis illustrations (Figure 3 and 4 panels C). Some additional explanations in either the legends or the text would help. There is not a clear conclusion I can learn from the paper which is really frustrating as the methodology seems to be sound and huge quantity of data was analyzed extensively (maybe overly tortured?).

Response 1: We thank the reviewer for their positive and valuable feedback.

We have added explanations for a better understanding of the PCA results in the captions of Figures 3 and 4, in the Methods section, line 191ff and in the Results section line 267ff.

As far as the supplementary material is concerned, all important statistical results are given in the main text. Most authors would leave it at that. In the spirit of full data availability, we have included the complete results of all statistical analyses for those interested in such details, but we do not believe that it is necessary to look at these tables at all to capture the results of the data analysis.

Finally, we completely rewrote the conclusion hoping to better summarize the obtained results.

Comment 2:  Minor:

Line 133: “19 D. suzukii…” should be “Nineteen D. suzukii…”

Line184: “variance method with the Euclidian Euclidean distance as a metric...."

Response 2: Thank you for finding these mistakes, both were corrected.

Reviewer 3 Report

Comments and Suggestions for Authors

The manuscript used calcium imaging to focus on the neural mechanism for the difference between Drosophila suzukii and D. melanogaster in response to fruit, leaf, and microbial odors. They found that some antennal lobe glomeruli differ in size and neuronal response to tested odors between the two species. The authors also assessed the behavioral responses to the tested odors but found no difference between the two species. Therefore, they concluded that the neural responses reflect species-specific shifts in the odor codes, but the neuronal responses may not directly reflect the different behavioral responses. They drew an appropriate conclusion based on their data. However, they should first examine whether these representative odors could reflect the behavioral responses to leaf, ripening, fermenting, and microbe. If they find the correct odorants, they will achieve the aim of the present study, how the first odor processing step in the antennal lobe diverges between species. Please provide the raw data about the results of calcium imaging.

Author Response

Comment 1: The manuscript used calcium imaging to focus on the neural mechanism for the difference between Drosophila suzukii and D. melanogaster in response to fruit, leaf, and microbial odors. They found that some antennal lobe glomeruli differ in size and neuronal response to tested odors between the two species. The authors also assessed the behavioral responses to the tested odors but found no difference between the two species. Therefore, they concluded that the neural responses reflect species-specific shifts in the odor codes, but the neuronal responses may not directly reflect the different behavioral responses. They drew an appropriate conclusion based on their data. However, they should first examine whether these representative odors could reflect the behavioral responses to leaf, ripening, fermenting, and microbe. If they find the correct odorants, they will achieve the aim of the present study, how the first odor processing step in the antennal lobe diverges between species. Please provide the raw data about the results of calcium imaging.

Response 1: We thank the reviewer for the positive feedback.

We had selected odours known to be emitted by leaves, ripening, fermenting, and microbial organisms. We agree with the reviewer that the conclusion from the lack of differences in behavioural responses must be that these odours are not the components that trigger the known divergence in host-seeking behaviour. Therefore, further efforts are needed in future studies to search for such active odorants. We have added this in the Discussion and Conclusion section, line 529ff and 546ff.

All raw data is available at https://github.com/NeurophysicsTrento/Comparative-Study-Drosophila, as we state in the Data Availability Statement.

Round 2

Reviewer 1 Report

Comments and Suggestions for Authors

The authors addressed the main review comments and improved the behavior figure. 

I still think the authors should discuss their choice of odors better for behavior and imaging. Hexanal seemed to be the most interesting odor in terms of glomerular response, was however not tested in behavior, the authors mention this now. 

Based on the new behavior figures, Isoamyl acetate did not elicit any significant preference in both species (compared to paraffin control - Figure 5d), thus, this might also not be the most relevant odor to use for calcium imaging. The authors should also comment on this problem.

I would suggest accepting the publication after these minor changes.

Author Response

Comment:

The authors addressed the main review comments and improved the behavior figure. 

I still think the authors should discuss their choice of odors better for behavior and imaging. Hexanal seemed to be the most interesting odor in terms of glomerular response, was however not tested in behavior, the authors mention this now. 

Based on the new behavior figures, Isoamyl acetate did not elicit any significant preference in both species (compared to paraffin control - Figure 5d), thus, this might also not be the most relevant odor to use for calcium imaging. The authors should also comment on this problem.

I would suggest accepting the publication after these minor changes.

Response:

We thank the reviewer for his positive evaluation of our work and we agree with his assessment that certain odours provide more relevant results than others. However, we would like to highlight that in this study the selection of odours was done at the beginning, based on prior knowledge and to ensure broad coverage of biologically relevant stimuli.

Adjusting the study design retrospectively based on final results is inconsistent with hypothesis-driven research and risks introducing bias. Even if odours yielded differing levels of relevance in our findings, all data was included to maintain transparency and reproducibility. Adding further odours after the complete statistical analysis, was not feasible under the same experimental conditions.

In lines 107ff, we provide details on the initial criteria for odour selection. In lines 439ff, 529ff, and 546ff we discuss the potential and necessity of follow-up studies to explore the behavioural relevance of further odours based on our current results.

We trust that this explanation will address the reviewer's concerns and reaffirm our commitment to rigorous and unbiased scientific practice.

Reviewer 3 Report

Comments and Suggestions for Authors

 I have no more comments on the manuscript.

Author Response

We thank the reviewer for his valuable contributions.